# Effective Model of Emerging Disease Prevention and Control in a High-Epidemic Area, Chiang Rai Province

**DOI:** 10.3390/ijerph22121849

**Published:** 2025-12-11

**Authors:** Jiraporn Sangsuwan, Phitsanuruk Kanthawee, Pamornsri Inchon, Phataraphon Markmee, Phaibun Chiraphatthakun

**Affiliations:** 1Department of Public Health, School of Health Science, Mae Fah Luang University, Chiang Rai 57100, Thailand; jiraporn.sang1995@gmail.com (J.S.); pamornsri.sri@mfu.ac.th (P.I.); phaibunchiraphatthakun@gmail.com (P.C.); 2Area-Based Research and Innovation in Cross-Border Health Care Group, School of Health Science, Mae Fah Luang University, Chiang Rai 57100, Thailand; 3Faculty of Public Health, Naresuan University, Phitsanulok 65000, Thailand; phataraphonm@nu.ac.th

**Keywords:** prevention management model, control of disease, border community, COVID-19

## Abstract

A concurrent mixed-methods study was conducted to examine the factors influencing COVID-19 prevention and control behaviors and to describe the management model implemented in Mae Sai District, a Thai–Myanmar border community, from June 2022 to May 2023. Mae Sai reported 21,890 confirmed cases and 12 deaths during the pandemic, underscoring the severity of the outbreak and the need for an effective local management model. Quantitative results indicated that attitudes, social support, participation, and service accessibility significantly influenced preventive behaviors among the general public. Among volunteers, perception and attitude were also significant, whereas only social support and participation were influential among government officials. The management model identified in this study demonstrated effectiveness through its coordinated multisectoral operations, high community compliance, and rapid cross-border communication. The model consisted of five components: emergency preparedness drills, organizational management through district and subdistrict disease control centers, a unified incident command system led by the district chief, coordinated domestic and international operations, and enforcement of control measures at formal checkpoints, natural crossings, and within communities. Successful implementation depended on strong collaboration among government agencies, volunteers, private organizations, local communities, and partners in Myanmar. This framework may serve as a practical guideline for managing other communicable diseases and enhancing preparedness for future health threats.

## 1. Introduction

COVID-19, a respiratory disease, was first identified in December 2019 in Wuhan, China. COVID-19 cases subsequently increased rapidly [1,2]. The World Health Organization declared the outbreak a pandemic on 11 March 2020, when cases exceeded 100,000 [2,3]. Within a few months, global cases reached 683 million, resulting in over 6.83 million deaths. In Thailand, the outbreak ranked 32nd globally, with 4.7 million confirmed cases, 4.6 million recoveries (97.9% of total cases), and 33,935 deaths [4]. Chiang Rai Province reported 174,842 confirmed cases and 83 deaths. Mae Sai District was among the most affected areas in the province, with 21,890 confirmed cases and 12 deaths [5]. The Thai government implemented stringent measures to monitor and control the spread of COVID-19.

Thailand established the Center for COVID-19 Situation Administration (CCSA) and declared a state of emergency [6,7,8]. Disease control centers were established at the provincial, district, and sub-district levels, assigning local administrations the responsibility for implementing government-mandated preventive measures to control the COVID-19 outbreak at both regional and national levels [9]. These measures included prohibiting or restricting activities involving close contact, regulating entry and exit from the country, and limiting mass movements across regions. Authorities also controlled vehicle use, traffic routes, goods, and medical supplies, and implemented work-from-home policies [10,11,12]. The Department of Disease Control and the Ministry of Public Health established COVID-19 prevention measures aligned with WHO guidelines, including the D-M-H-T-T-A measures [13,14,15]. However, an effective response to the COVID-19 outbreak requires cooperation from all stakeholders, despite the existence of these measures and guidelines at national and provincial levels.

Research on individual preventive behaviors has identified several determinants using the PRECEDE model [16]. Predisposing factors motivate desired behaviors, including attitudes and health perceptions. Enabling factors facilitate behavior by providing access to resources and services, while reinforcing factors involve external influences such as social support and participation [17,18]. Among health personnel, preventive behaviors are shaped by knowledge, willingness to act [19], and appropriate training in the use of protective equipment [20]. Strengthening the knowledge and understanding of health teams is essential for effective patient care and disease prevention. However, existing studies have mainly focused on general communities or specific occupational groups, with limited methodological evidence on how multilevel behavioral determinants function within complex border populations. There remains a lack of research that integrates behavioral factors with real-world management structures in high-mobility border settings, creating an important methodological gap. Mae Sai District, which borders Tachileik, Myanmar [21], experienced high COVID-19 transmission due to intense population movement through formal and informal routes, including international checkpoints, relaxed crossing areas, and natural paths. Beyond the high case burden, Mae Sai is characterized by multiple entry channels, ethnic and linguistic diversity, and frequent undocumented migration, all of which contribute to variation in information access, risk perception, and preventive practices among residents. These contextual factors influenced the effectiveness of local disease prevention and control systems [22]. Additionally, the POCCC framework (Planning, Organization, Coordination, Command, and Control) is introduced to guide the analysis of management structures in this high-risk border setting.

## 2. Materials and Methods

### 2.1. Study Design and Study Setting

A concurrent mixed-methods approach was employed, with quantitative and qualitative data collected and analyzed simultaneously. Quantitative data were obtained through structured questionnaires to investigate predisposing, reinforcing, and enabling factors influencing COVID-19 surveillance, prevention, and control. Qualitative data were collected through in-depth interviews and focus group discussions to explore the management model for COVID-19 prevention and control in the border community of Mae Sai District, Chiang Rai Province.

Integration of the quantitative and qualitative components was carried out through methodological triangulation. Quantitative results were compared with qualitative themes to validate findings across data sources, enhance interpretive accuracy, and provide a more comprehensive understanding of the underlying issues. This triangulated approach enabled each method to complement the others, offering distinct and in-depth insights and providing a multifaceted perspective on the identified challenges.

The study area was selected using purposive sampling, specifically targeting the Mae Sai Subdistrict in Mae Sai District due not only to its high cumulative number of COVID-19 cases but also to its unique characteristics as an international border community with multiple formal and informal entry points, high cross-border mobility, ethnic diversity, and complex population movements all of which make it a high-risk setting requiring context-specific investigation.

Chiang Rai, Thailand’s northernmost province, shares borders with Myanmar’s Shan State to the north, Laos’ Bokeo Province to the east, Phayao to the south, Lampang to the southwest, and Chiang Mai to the west. The province is linked to Houayxay, Laos, via the Fourth Thai–Lao Friendship Bridge, spanning the Mekong River [23]. Mae Sai Subdistrict, situated at Chiang Rai’s northernmost point, directly borders Myanmar and is approximately 63 km from Mueang Chiang Rai District. Mae Sai District, comprising eight subdistricts, serves as a key border crossing between Thailand and Tachileik, Myanmar. It features two permanent border checkpoints at Mae Sai Bridges 1 and 2, along with five border trade checkpoints in Wiang Phang Kham, Mae Sai, and Ko Chang Subdistricts, facilitating trade and cross-border movement (as shown in Figure 1).

Mae Sai is home to a diverse ethnic population, including the Akha, Hmong, Lahu (Musoe), Chinese Haw (Yunnanese Chinese), Tai Ya, Dara-ang, Lua, Tai Khuen, Tai Yong, and Shan (Tai Yai) communities, each contributing to the region’s rich cultural heritage.

### 2.2. Sample Size Calculation

Quantitative Research Sample Levels:

The population and sample were divided into three levels:

Level 1: General Public

The sample size was calculated using W.G. Cochran’s formula [24], and there were 346 participants in total.Proportionate stratified random sampling was used to ensure an even distribution of participants from each village based on population proportions.The sample from each village was selected using probability sampling through simple random sampling by drawing lots from a list of target individuals, following inclusion criteria: individuals aged 20 and above, residing in Mae Sai Subdistrict for at least one year, willing to participate in the study, and able to communicate in Thai.

Level 2: Volunteer Workers

2.1 Village Health Volunteers and Migrant Health Volunteers: The sample size was calculated using W.G. Cochran’s formula [25], resulting in 197 participants.
○Inclusion criteria: volunteers with at least six months of experience in COVID-19 surveillance, prevention, and control who are willing to participate in the study.2.2 Community Leaders, Territorial Defense Volunteers, Civil Defense Volunteers, and Village Security Teams: These were selected through purposive sampling, with a total of 36 participants.
○Inclusion criteria: volunteers with at least six months of experience in COVID-19 surveillance, prevention, and control who are willing to participate in the study.

Level 3: Government Officials in COVID-19 Surveillance and Control in the Border Area of Mae Sai District, Chiang Rai Province

Selected through purposive sampling with the following inclusion criteria: officials with at least six months of experience in COVID-19 surveillance, prevention, and control in Mae Sai District and willing to participate in the study.
○3.1 Health Officials in Mae Sai Subdistrict Health Promoting Hospital: 10 participants.○3.2 Local Government Officials in Mae Sai Subdistrict: 10 participants.○3.3 Security Officials in the area, including immigration police and military personnel: 10 participants.

Qualitative Research Sample Levels:

The qualitative sample was selected through purposive sampling:

Level 1: General Public

There were 12 participants (one person from each of the 12 villages, with each village having one Thai representative and one migrant representative).

Level 2: Volunteer Workers

Total of 12 participants:
○Three Village Health Volunteers.○Three Migrant Health Volunteers.○Three Community Leaders.○One Territorial Defense Volunteer.○One Civil Defense Volunteer.○One Village Security Team Member.

Level 3: Government Officials

Total of 15 participants:
○Five Health Officials from the Subdistrict Health-Promoting Hospital.○Four Local Government Officials from Mae Sai Subdistrict.○Two Immigration Police Officers.○Three Military Personnel.

### 2.3. Research Instruments

A structured and validated questionnaire comprising five parts was utilized to collect information. Part I gathered personal data, including sociodemographic information, history of COVID-19 infection, and vaccination status. Part II focused on predisposing factors, with questions related to perception (10 items) and attitude (10 items); part iii addressed reinforcing factors, covering social support (10 items) and participation (10 items). Part IV examined enabling factors, consisting of questions related to services (10 items). Part V focused on COVID-19 prevention and control behavior (10 items). Because the study population included ethnic minority and migrant groups, the questionnaire was administered in Thai with assistance from trained bilingual interpreters when needed to ensure comprehension and accurate responses.

The questionnaire was created by the researcher, the content validity was assessed by three experts, and the content validity index (CVI) method was used to improve the content validity. Reliability was also examined by administering the tool to a sample of 30 individuals in Pongpha Subdistrict, Mae Sai District, Chiang Rai Province, which had characteristics similar to those of the target population. The Kuder–Richardson coefficient (KR-20) was 0.73, which is acceptable for exploratory research but is considered borderline; therefore, the reliability should be interpreted with caution.

### 2.4. Data Analysis

Statistics were used for quantitative data analysis. Descriptive statistics were computed for all sections of the questionnaires using SPSS version 23. Categorical variables are presented as frequencies and percentages. Composite scores for variables in Parts II–V were converted to percentage scores and classified into three levels according to Bloom’s cutoff criteria: low (<60%), moderate (60–79%), and high (80–100%) [26]. Spearman rank correlations were used to detect the correlation at the significant level of alpha = 0.05. Multiple regression analysis was also performed to identify predictors of COVID-19 prevention and control behaviors.Qualitative data were analyzed using a conventional content analysis approach. Line-by-line open coding was conducted on the interview and focus group transcripts, after which similar codes were grouped into categories and developed into higher-order themes aligned with the study framework. To enhance analytical rigor, two researchers independently coded a subset of transcripts, and coding discrepancies were discussed and resolved through a consensus-based process. Data saturation was considered to have been reached when no new codes or themes emerged across successive interviews and focus groups. Methodological triangulation was undertaken by comparing themes across participant levels (general public, volunteers, and government officials) and by cross-validating qualitative findings with quantitative results.

### 2.5. Ethics Consideration

This study and its protocols were approved by the Human Research Ethics Committee of the Chiang Rai Provincial Public Health Office (CRPPHO No. 94/2565) and the Human Research Ethics Committee of Mae Fah Luang University (No. COA139/2022). All participants were clearly informed of the study objectives, procedures, confidentiality measures, potential risks, and benefits prior to participation. Written informed consent was obtained after ensuring that participants fully understood their rights, including the right to refuse or withdraw at any time without penalty.

Additional safeguards were implemented to ensure adequate protection of vulnerable populations, including older adults, migrant workers, and individuals with limited literacy. Participation was strictly voluntary, and information was provided using clear, culturally appropriate language. Data collection took place in safe, non-coercive community settings. For participants with low literacy, trained interviewers assisted in reading and interpreting questionnaire items to enhance comprehension while maintaining autonomy and confidentiality. No personally identifiable information was collected, thereby minimizing risk and ensuring the privacy and protection of all participants.

## 3. Results

### General Characteristics of Participants

The majority of the general public level were female (59.8%), aged between 30–39 and 40–49 years (22.2%), and married (53.2%). Most participants were Buddhists (91.9%), had a primary education level (32.9%), and were covered by the Universal Coverage Scheme (67.9%). The predominant occupation was daily wage earners (50.0%). A significant portion had a history of COVID-19 infection (76.6%), with many undergoing home isolation (HI) (39.0%). Additionally, the majority had received two doses of the COVID-19 vaccine (52.3%) (Table 1).

Among the volunteer workers, the majority were male (51.5%), aged between 30 and 49 years (43.3%), and married (74.2%). Most were Buddhists (94.8%), had a primary education level (39.9%), and were covered by the Universal Coverage Scheme (82.8%). The predominant occupation was daily wage earners (55.8%). A significant portion had a history of COVID-19 infection (79.0%), with many undergoing home isolation (HI) (39.5%). Additionally, the majority had received three doses of the COVID-19 vaccine (54.1%) (Table 1).

For the government officials, the majority were female (60.0%), aged between 30 and 39 years (36.7%), and single (53.3%). Most were Buddhists (86.8%), had a bachelor’s degree (66.7%), and were covered by civil servant medical benefits (33.3%). The predominant occupation was government officials (66.7%). A significant portion had a history of COVID-19 infection (70.0%), with many undergoing home isolation (HI) (43.3%). Additionally, the majority had received four doses of the COVID-19 vaccine (66.7%) (Table 1).

Predisposing Factors: Across all three levels, perception consistently showed a high level, with 95.4% of community members, 87.1% of volunteers, and 93.3% of government officials scoring in the high range. In contrast, attitude levels displayed more variation. The community members predominantly demonstrated a high attitude level (93.4%), whereas volunteers showed a primarily moderate level (48.9%), and government officials also presented a moderate level (70.0%). This pattern suggests that while foundational understanding of COVID-19 was strong across levels, attitudinal readiness varied, particularly among volunteers and officials (Table 2).

Reinforcing Factors: Social support for the general public mainly was high (88.2%); for volunteer workers it was high (86.7%); and for government officials it was low (53.3%). Participation levels were high for the general public (73.7%), high for volunteer workers (57.1%), and moderate for government officials (36.7%). (Table 2)

Enabling Factors: Access to services was mainly high for the general public (81.8%), high for volunteer workers (62.7%), and high for government officials (50.0%) (Table 2).

COVID-19 Prevention and Control Behaviors: Prevention and control behaviors were mainly high among the general public (89.6%), high among volunteer workers (84.5%), and high among government officials (70.0%) (Table 2).

General Public: The predisposing factor of attitude has a significant positive correlation with COVID-19 prevention and control behaviors (r = 0.14, *p*-value = 0.008). The reinforcing factor of social support has a significant positive correlation with COVID-19 prevention and control behaviors (r = 0.66, *p*-value < 0.001). The reinforcing factor of participation significantly correlates positively with COVID-19 prevention and control behaviors (r = 0.61, *p*-value < 0.001). The enabling factor of service systems has a significant positive correlation with COVID-19 prevention and control behaviors (r = 0.65, *p*-value < 0.001) (Table 3).

Volunteer Workers: The predisposing factor of perception significantly correlates positively with COVID-19 prevention and control behaviors (r = 0.18, *p*-value = 0.007). The predisposing factor of attitude significantly positively correlates with COVID-19 prevention and control behaviors (r = 0.26, *p*-value < 0.001). The reinforcing factor of social support has a significant positive correlation with COVID-19 prevention and control behaviors (r = 0.38, *p*-value < 0.001). The enabling factor of service systems has a significant positive correlation with COVID-19 prevention and control behaviors (r = 0.16, *p*-value = 0.015) (Table 3).

Government Officials: The reinforcing factor of social support has a significant positive correlation with COVID-19 prevention and control behaviors (r = 0.50, *p*-value = 0.005). The reinforcing factor of participation significantly correlates positively with COVID-19 prevention and control behaviors (r = 0.47, *p*-value = 0.008) (Table 3).

Qualitative Data on the COVID-19 Prevention and Control Management Model in the Border Community of Mae Sai District, Chiang Rai Province.

The results synthesized using the POCCC framework are presented in Table 4.

Planning: Preparedness before the outbreak involved creating a three-phase management plan: pre-event, event, and post-event. This included conducting drills with government agencies, the private sector, local administrative organizations, charitable organizations, health officials, village heads, health volunteers, and migrant volunteers. Training and development of personnel were conducted to prepare for public health emergencies, including establishing a coordination system with the communication network, as illustrated by the interview transcripts. 

“In the early phase, we focused on preparedness by ensuring resources and preventive measures were in place, including monitoring the situation and educating the public on protective measures.”(Local Administrative Officer 1, 29 October 2022, Interview No. 04)

“A simulation exercise involved government agencies, private organizations, local administrative bodies, charitable organizations, public health personnel, village heads, and health volunteers—totaling 200 participants.”(International Disease Control Officer 2, 28 October 2022, Interview No. 06)

“More than 200 personnel from over 70 CDCU teams across the province were trained in public health emergency response. We also developed a comprehensive data system to manage emergencies effectively.”(Subdistrict Health Promoting Hospital Officer 1, 27 September 2022, Interview No. 01)

Organizing: The area’s surveillance, prevention, and control system was managed according to the orders of the Chiang Rai Provincial Communicable Disease Committee. Implementation was carried out through the district and sub-district disease control centers and the emergency response centers for public health. Policies and measures from the provincial committee were executed at various levels, with border command centers coordinating and integrating the efforts of multiple parties to address border control issues in Line with the directives of the provincial committee and local border committees. Coordination was carried out with local security agencies in neighboring countries to analyze and resolve the COVID-19 outbreak at the border and provincial levels. Emergency operations centers at international entry points controlled and responded to outbreaks among travelers and transport operators.

“At the organizational level, the Provincial Communicable Disease Committee was the first to be formed, followed by the establishment of a district-level Emergency Operations Center (EOC), led by the Director of Mae Sai Hospital and the District Public Health Officer, with supporting committees structured under the Public Health Incident Command System.”(International Disease Control Officer 1, 28 October 2022, Interview No. 07)

“The EOC is the core of emergency response, functioning as a shared operational space for various agencies under the incident command system. It facilitates decision-making, coordination, and the rapid exchange of information and resources during the COVID-19 outbreak.”(Subdistrict Health Promoting Hospital Officer 1, 27 September 2022, Interview No. 02)

Commanding: At the district level, there was a unified command system with the Mae Sai district chief as the sole incident commander, who maintained good relations with local leaders in the neighboring country. At the sub-district level, operations were conducted through the Mae Sai Sub-district Disease Control Center, led by the deputy district chief responsible for Mae Sai. The command team included police from the Mae Sai Police Station, immigration police, special task forces from the 3rd Cavalry Regiment, Pha Muang Force, local administrative officials from Mae Sai Subdistrict Municipality and Mae Sai Friendship Subdistrict Municipality, schools, and other government agencies in the area, as well as village heads and village health volunteer leaders, with the director of the Mae Sai Subdistrict Health Promoting Hospital acting as the secretary.

“The District Chief Officer acted as the Incident Commander at the district level, demonstrating strong leadership, experience, and commitment to problem-solving. With multiple committees and agencies involved, communication was clear, fostering mutual understanding and reducing confusion in implementing orders.”(Local Administrative Officer 3, 29 October 2022, Interview No. 05)

Coordinating: International coordination between Mae Sai, Thailand, and Tachileik, Myanmar, was carried out formally through TBC and informally through on-site officers. Local coordination in the Mae Sai Subdistrict was conducted through the structure of the district and sub-district disease control centers and the emergency medical and public health operations center. Coordination was managed from top to bottom, bottom to top, or laterally through meetings and official documents, although communication through these channels could be faster. Another coordination method was through personal relationships within and between agencies, using direct communication, Line, and Facebook, which could provide faster and more timely communication depending on acceptance and understanding among the informal groups involved.

“Information was mainly shared via LINE application. When a COVID-19 case was detected, it was reported from the Mae Sai Hospital outbreak center to the local health-promoting hospital (HPH), which then informed community leaders and village health volunteers. The HPH also reported cases to the Subdistrict Disease Control Operations Center to monitor patients and quarantined individuals, allowing local administrative organizations to provide necessary support.”(Subdistrict Health Promoting Hospital Officer 1, 27 September 2022, Interview No. 01)

“There was no direct coordination with Myanmar due to the border closure. The International Communicable Disease Control Unit handled communication. In cases of illegal border crossings, military forces were responsible for apprehension and notified the police, who then informed Immigration and District Health Authorities for disease screening. Law enforcement dealt with arrests and deportations, except for Thai nationals, who were subject to fines or legal action.”(Subdistrict Health Promoting Hospital Officer 1, 27 September 2022, Interview No. 03)

Controlling: Control measures were implemented through four main channels: (1) international checkpoints, (2) relaxed points, (3) natural routes, and (4) within the community. These measures were carried out by local communicable disease control officers, community leaders, village health volunteers, border community health volunteers, civil defense volunteers, village security teams, immigration police, and military personnel, strengthening the local system.

“There were multiple channels for enforcing control measures. In Mae Sai, the four main channels included international checkpoints managed by the Disease Control Unit and Immigration, border trade checkpoints (which were temporarily closed), natural border crossings, which posed the biggest challenge due to frequent illegal crossings, and community-level monitoring handled by local health authorities and community networks.”(International Disease Control Officer 2, 28 October 2022, Interview No. 07)

“For event control measures, organizers had to seek district approval and document compliance, reporting to the local health unit and the Subdistrict Disease Control Operations Center.”(Subdistrict Health Promoting Hospital Officer 2, 27 September 2022, Interview No. 01)

The COVID-19 prevention and control management model for the Thai–Myanmar border area in Mae Sai District, Chiang Rai Province, employed international cooperation mechanisms through TBC. The country’s management was divided into three levels: provincial, district, and sub-district, through the disease control operations centers, managed according to the POCCC model. The key feature was the unified command system with a single commander, reducing confusion in operations. The area’s operations integrated efforts from all sectors, including government officials from health, security, administrative agencies, local administrative organizations, village health volunteers, migrant health volunteers, and the local community. Predisposing, enabling, and reinforcing factors influenced the prevention and control efforts of officials, volunteers, and the public. Operations were monitored and evaluated, with results summarized and reported at district and sub-district disease control center meetings to discuss, analyze, and jointly resolve issues (as shown in Figure 2 and Figure 3).

**Figure 2 ijerph-22-01849-f002:**
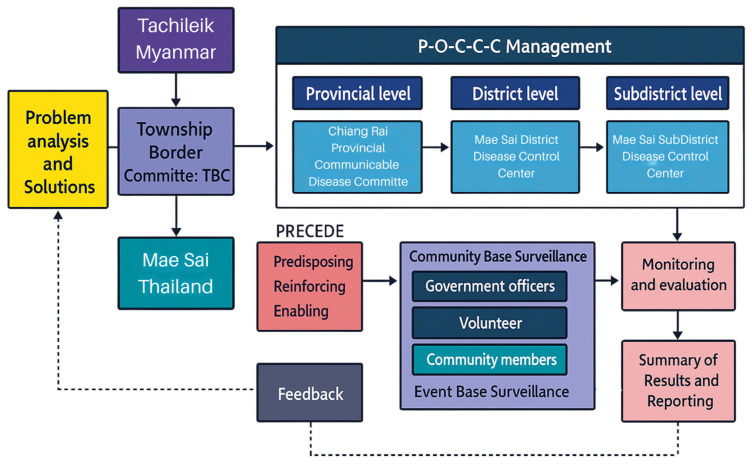
The Management Model for Prevention and Control of COVID-19 in the Border Area between Thailand and Myanmar, Mae Sai District, Chiang Rai Province.

**Figure 3 ijerph-22-01849-f003:**
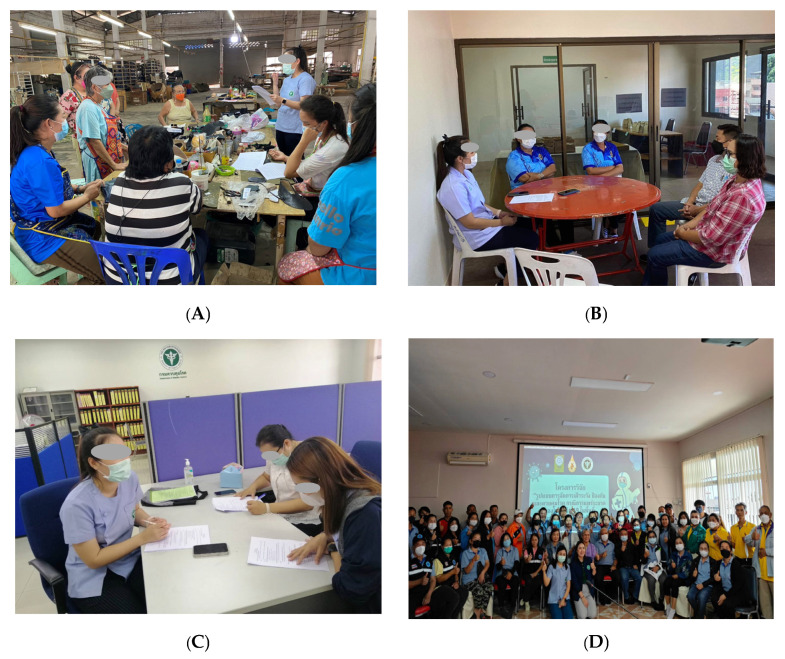
Field Data Collection Activities. (**A**) Household interviews with community members in the border area. (**B**) In-depth interviews with volunteer health workers at the subdistrict health center. (**C**) Key informant interviews with government officials at the International Communicable Disease Control Checkpoint. (**D**) Stakeholder workshop and community engagement meeting involving public health personnel, volunteers, and community leaders.

## 4. Discussion

General Public

Attitude: There was a positive correlation between attitude and COVID-19 prevention and control behaviors. Qualitative findings indicated that the general public received information from the COVID-19 Situation Administration Center through television and social media, as well as from health officials, health volunteers, and community leaders within the community. This information led to a positive change in attitude towards self-protection and cooperation in preventing and controlling the spread of COVID-19. This finding aligns with Nattawan Khamsen’s study, which showed that participants were confident that COVID-19 would eventually be controlled and believed Thailand would overcome the pandemic. This confidence was attributed to the comprehensive and daily information provided by the COVID-19 Situation Administration Center, which emphasized trust in government efforts to control the outbreak [28]. Similarly, a study on Chinese citizens’ knowledge, attitudes, and practices during the COVID-19 outbreak found that most participants believed China would ultimately overcome the pandemic [29].

Social Support: There was a positive correlation between social support and COVID-19 prevention and control behaviors. Qualitative findings revealed that the public received survival kits from local administrative organizations, the private sector, and local residents. Information about COVID-19 was disseminated through social media, health officials, village health volunteers, and community leaders, who provided knowledge and recommendations on behavior modification to avoid risk factors. Social support from family, friends, health volunteers, and health personnel motivated individuals to change their behaviors [30,31].

Service Systems: A positive correlation existed between service systems and COVID-19 prevention and control behaviors. Qualitative findings showed that proactive efforts by health officials in the community, such as educating the public on self-protection, providing care based on symptoms, and distributing medications through village health volunteers, were practical. This approach aligns with the World Health Organization and Western Pacific Region’s recommendation for primary health care services to deliver medications to patients’ homes and extend service hours to reduce congestion at health facilities [32].

2.Volunteer Workers

Perception: Perception was positively correlated with COVID-19 prevention and control behaviors. This finding is consistent with Salakham et al.’s study, which found that the perception of disease severity was positively correlated with the performance of COVID-19 control tasks, including symptom screening, monitoring, surveillance, reporting, and overall performance [33]. Similarly, Kanjana Panyathorn et al. found that village health volunteers played a significant role in COVID-19 prevention and control efforts, working closely with health officials and related agencies [34]. Kittiporn Naosuwan et al. also found that the perception of disease severity was positively correlated with the role of village health volunteers in community COVID-19 control efforts [35].

Attitude: A positive correlation existed between attitude and COVID-19 prevention and control behaviors. This finding aligns with Siuki et al.’s research, which showed that health education on HIV and AIDS prevention behaviors was correlated with attitudes, subjective norms, and perceived behavioral control among health volunteers in Iran [36]. Supanyabut’s study also found that attitudes influenced the correct implementation of influenza A (2009 H1N1) prevention behaviors, suggesting that knowledge and attitudes are crucial for proper behavior [37].

Social Support: There was a positive correlation between social support and COVID-19 prevention and control behaviors. Yuthana Yaibkai’s study found that social support positively impacted the performance of village health volunteers in Sukhothai Province, as support from family, neighbors, government, and private organizations provided encouragement, tools, coordination, and experience sharing [38]. Supaporn Wongthi’s study also found that social support was a factor in COVID-19 prevention efforts, as family, friends, the community, and health officials encouraged village health volunteers to model good preventive behaviors [39].

Service Systems: A positive correlation existed between service systems and COVID-19 prevention and control behaviors. Effective prevention and control services, such as disease investigation, community surveillance networks, and risk reduction for non-communicable diseases, involved proactive health services integrated with community networks through information technology [40]. During the COVID-19 outbreak, health officials coordinated informally with village health volunteers through Line, which aligns with Cholthicha Chum-In’s study. The Ban Klong Muan Subdistrict Health Promoting Hospital maintained diverse communication channels, including public announcements, Line, and Facebook, to provide information on COVID-19 [41]. Pietig recommended that health service providers use multiple communication channels to maintain contact and provide accurate information during the COVID-19 crisis [42].

3.Government Officials

Social Support: There was a positive correlation between social support and COVID-19 prevention and control behaviors. Qualitative findings indicated that government agencies received support for COVID-19 prevention materials from private organizations and local foundations. This finding aligns with Phiphob Janmuean’s study, which found that social support factors, including the availability of protective equipment, significantly impacted prevention behaviors among health personnel, as protective equipment from the Ministry of Public Health and donations from the public and other agencies were perceived as sufficient [43]. However, this contrasts with previous studies by Houghton et al. and Chutima Deesawat et al., which found that access to and support for personal protective equipment influenced the willingness to follow respiratory infection prevention and control guidelines [44,45].

Participation: A positive correlation existed between participation and COVID-19 prevention and control behaviors. Qualitative findings showed that health officials coordinated with various agencies under the Mae Sai district/subdistrict disease control operations center, aligning with Kasemsook Kanchaiyaphum’s study. This study found that most health officials participated in activities based on public health emergency response plans, including COVID-19 control efforts, which required the involvement of all sectors, from problem analysis to joint activities through the emergency operations centers, leading to faster outbreak control and reduced community anxiety [46].

Surveillance, Prevention, and Control Model: The COVID-19 surveillance, prevention, and control model in Mae Sai District, Chiang Rai Province, involved formal and informal international cooperation through the local border committee (TBC). This finding aligns with Saowanee Plienpanich et al.’s study, which highlighted the informal working methods of Thai–Lao border public health operations that facilitated policy implementation at the local level despite formal limitations [47]. The country’s management was divided into provincial, district, and sub-district levels through disease control operations centers, following the POCCC model and emphasizing community participation. The integration of efforts from all sectors, including health, security, administrative agencies, local governments, village health volunteers, migrant health volunteers, and the local community, was facilitated through electronic communication for rapid information sharing. This approach aligns with the COVID-19 surveillance, prevention, and control model at Udonthani Subdistrict Health Promoting Hospital, which involves effective surveillance, screening, investigation, and control; capacity building for officials; accurate and rapid information dissemination to volunteers and the community; and participation from all relevant sectors. The success factors included strong involvement from the district health development committee and network partners in the “Udon People Do not Abandon Each Other, Together We Overcome COVID” initiative, and the use of modern technology for rapid information dissemination [48]. This approach also aligns with Phuchong Chuenchom et al.’s study, which found that integrating ideas from multiple agencies led to appropriate solutions and to unified management under a single director or authority figure and a single communication channel [49]. Kessorn Thaononiw’s research emphasized building disease surveillance networks involving various agencies, leading to apparent task execution and enhanced cooperation, which could be expanded to neighboring countries [50]. Finally, Rungreung Kitpati et al. found that community participation was crucial in responding to the COVID-19 crisis [51].

This model has potential applicability beyond the COVID-19 context. The PRECEDE behavioral determinants, which include predisposing factors, enabling factors, and reinforcing factors, together with the POCCC management framework, can be adapted to other emerging infectious diseases such as dengue, influenza, and acute respiratory infections. Because these components emphasize strengthening risk perception, community participation, service accessibility, and coordinated multisectoral response, the model can enhance preparedness and response capacities during future outbreaks, particularly in border and high-risk settings.

The findings indicate that the Mae Sai COVID-19 prevention and control model was effective, demonstrated by the high levels of preventive behaviors across levels—89.6% among the general public, 84.5% among volunteers, and 70.0% among government officials. Epidemiological reports also showed fewer community transmission clusters compared with other border districts such as Mae Sot during the same period [52,53]. This supports evidence that multi-layered, community-based systems integrating surveillance, coordination, and community participation are effective in high-mobility border settings [54,55].

Differences in significant predictors across levels reflect their roles and exposure to health information. For the general public, reinforcing and enabling factors (social support, participation, and service systems) were most influential, consistent with studies showing that community behavior depends heavily on external support and access to services [56,57]. Among volunteers, perception and attitude were significant due to their training and continuous engagement in surveillance activities, aligning with findings that frontline volunteers’ behaviors are shaped by knowledge-driven processes [58,59]. Government officials showed significance only in reinforcing factors, likely due to uniformly high baseline knowledge and structured operational responsibilities [60]. The small sample size (*n* = 30) for this level may also limit power and increase the likelihood of Type II error. Benchmark comparisons with Mae Fah Luang, Chiang Khong, and Mae Sot demonstrate that Mae Sai’s stronger integration of migrant volunteers and use of rapid informal communication channels contributed to more stable outbreak patterns [52,53,54,55,61]. While the model shows potential for transferability, contextual adaptation is still required for areas with different migration dynamics and administrative structures.

The findings of this study also offer important implications for public health policy and future communicable disease preparedness. The strong influence of social support, participation, and service accessibility highlights the need to reinforce community-based surveillance systems, especially in border areas with dynamic population mobility. Policymakers should prioritize cross-sectoral coordination involving health authorities, local administrative organizations, security agencies, and volunteer networks to ensure rapid information flow and coherent operational responses. The unified command structure implemented in Mae Sai illustrates a practical approach that could be institutionalized within national preparedness strategies.

Moreover, the significant roles played by volunteers and migrant networks underscore the importance of investing in training, resource allocation, and risk communication that is tailored to diverse populations. Strengthening binational coordination mechanisms, such as local border committees, will be essential for managing future outbreaks with cross-border transmission potential. These insights can guide the development of preparedness plans not only for COVID-19 but also for other emerging and re-emerging infectious diseases in border regions.

## 5. Limitations

Although the study included three stakeholder levels (community members, volunteer workers, and government officials), the sample sizes across these levels were not balanced, which limited the ability to compare differences between levels.The study was unable to perform multivariate regression to adjust for confounders due to small and unbalanced sublevel sizes, particularly the limited number of government officials (*n* = 30), which may leave some residual confounding.Older adults, who represent a key high-risk population for COVID-19, were underrepresented in the sample, which may reduce the applicability of the findings to this age group.Although a mixed-methods approach was employed, the quantitative sample sizes for some sublevels remained limited, potentially affecting statistical power and the stability of correlation estimates.The cross-sectional design and reliance on self-reported data restrict causal interpretation and may introduce recall and social desirability biases in reporting preventive behaviors.Purposive sampling and the focus on a single border district limit the generalizability of the findings to other regions with different sociocultural or migration contexts.

## 6. Conclusions

The findings from this study provide important implications for strengthening epidemic preparedness and response in border communities. Differences in the determinants associated with COVID-19 prevention and control behaviors among the general public, volunteers, and government officials demonstrate that risk communication and behavioral strategies must be tailored to level-specific drivers. Reinforcing and enabling factors were most influential for the general public, perception- and attitude-based factors played a stronger role for volunteers, and reinforcing mechanisms were particularly important for government officials. These variations underscore the need for differentiated public health interventions that align with the functional responsibilities, exposure to information, and behavioral motivations of each level.

The integrated surveillance, prevention, and control model in Mae Sai offers a structured and practical framework for managing infectious disease threats in border settings. Core components such as unified command, multisectoral coordination, risk communication, and service-system support appear transferable to other high-mobility districts with similar administrative and logistical structures. However, certain elements—including migrant–volunteer engagement, reliance on informal cross-border communication, and community participation specific to Mae Sai’s sociocultural context—are context-dependent and may require adaptation when applied elsewhere. These distinctions help clarify both the generalizability and limitations of the model.

Based on the quantitative patterns observed, prioritized recommendations include enhancing social support mechanisms and improving access to health services for the general public, strengthening perception- and attitude-based capacity-building programs for volunteers, and reinforcing participation and community-engagement channels within government agencies. Future research should consider longitudinal designs to monitor behavioral change over time, validate the model in other border regions with varying migration dynamics, and incorporate objective indicators such as real-time surveillance timeliness, mobility data, or verified compliance measures to strengthen empirical robustness and guide broader model adoption.

## Figures and Tables

**Figure 1 ijerph-22-01849-f001:**
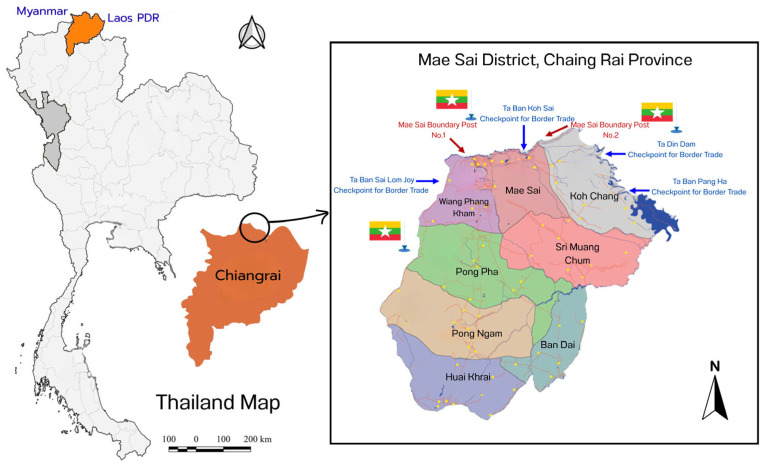
Map of Subdistricts in Mae Sai District, Chiang Rai Province, Thailand. Note: The yellow dots on the map represent village locations (individual villages/settlements) within each subdistrict of Mae Sai District.

**Table 1 ijerph-22-01849-t001:** General characteristics of participants.

Characteristics	Community Member		Volunteer		Government Officials	
	*n*	%	*n*	%	*n*	%
Total	346	100.0	233	100.0	30	100.0
Sex						
Male	139	40.2	120	51.5	12	40.0
Female	207	59.8	113	48.5	18	60.0
Age (years)						
20–29	58	16.8	0	0.0	8	26.7
30–39	76	22.0	21	9.0	11	36.7
40–49	76	22.0	80	34.3	5	16.5
50–59	98	28.3	93	39.9	6	20.0
≥60	38	11.0	39	16.7	0	0.0
Marital status						
Single	127	36.7	36	15.5	16	53.3
Married	184	53.2	173	74.2	12	40.0
Widowed/Divorced	26	7.5	9	3.9	0	0.0
Religion						
Buddhist	318	91.9	221	94.8	26	86.7
Christian	17	4.9	12	5.2	3	10.0
Islamic	10	2.9	1	0.4	1	3.3
Unreligious	1	0.3	0	0.0	0	0.0
Education						
Unlettered	90	26.0	23	9.9	1	3.3
Elementary School	114	32.9	93	39.9	3	10.0
Secondary School	39	11.3	40	17.2	20	66.7
High School	52	15.0	57	24.5	6	20.0
Bachelor’s degree	34	9.8	20	8.6	1	3.3
Master’s Degree	6	1.7	0	0.0	3	10.0
Medical treatment rights						
Universal Coverage Scheme	235	67.9	193	82.8	0	0.0
Social Security Fund	36	10.4	18	7.7	10	33.3
Stateless People	34	9.8	13	5.6	0	0.0
No rights	30	8.7	0	0.0	0	0.0
Civil Servant Medical Benefit Scheme	11	3.2	9	3.9	20	66.7
Occupation						
Employee	173	50.0	130	55.8	0	0.0
Merchant	63	18.2	40	17.2	0	0.0
Agriculturist	27	7.8	40	17.2	0	0.0
Housekeeper	24	6.9	15	6.4	0	0.0
Government Officer	13	3.8	1	0.4	20	66.7
Government Employee	4	1.2	0	0	10	33.3
Self-Employed.	6	1.7	1	0.4	0	0.0
State Enterprise	3	0.9	4	1.7	0	0.0
Unemployed	17	4.9	2	0.9	0	0.0
Students	13	3.8	0	0	0	0.0
Freelance	1	0.3	0	0	0	0.0
History of COVID-19 infection						
Yes	265	76.6	184	79.0	21	70.0
No	81	23.4	49	21.0	9	30.0
Place of treatment						
Chiangrai Prachanukroh Hospital	34	12.8	8	3.4	1	3.3
Mae Sai Hospital	62	17.9	56	24.0	4	13.3
Kasemrad Sriburin Hospital Mae Sai	8	2.3	17	7.3	2	6.7
Community isolation (CI)	26	7.5	11	4.7	1	3.3
Home isolation (HI)	135	39.0	92	39.5	13	43.3
History of vaccination						
Yes	335	96.8	231	99.1	30	100.0
No	11	3.2	2	0.9	0	0.0
Number of vaccinations (Dose)						
1	4	1.2	11	4.8	0	0.0
2	181	52.3	60	26.0	1	3.3
3	140	40.5	125	54.1	9	30.0
4	10	2.9	35	15.2	20	66.7

**Table 2 ijerph-22-01849-t002:** Level of predisposing, enabling, reinforcing, and COVID-19 prevention and control behaviors.

Factors	Community Member (*n* = 346)		Volunteer (*n* = 233)		Government Officials (*n* = 30)	
	*n*	%	*n*	%	*n*	%
Independent variables						
Predisposing						
Perception						
Low	6	1.7	10	4.3	0	0.0
Moderate	10	2.9	20	8.6	2	6.7
High	330	95.4	203	87.1	28	93.3
Attitude						
Low	5	1.4	54	23.2	4	13.3
Moderate	18	5.2	114	48.9	21	70.0
High	323	93.4	65	27.9	5	16.7
Reinforcing						
Social support						
Low	28	8.1	19	8.2	16	53.3
Moderate	13	3.8	12	5.2	4	13.3
High	305	88.2	202	86.7	10	33.3
Participation						
Low	34	9.8	55	23.6	9	30.0
Moderate	57	16.5	45	19.3	11	36.7
High	255	73.7	133	57.1	10	33.3
Enabling						
Service						
Low	21	6.1	24	10.3	4	13.3
Moderate	42	12.1	63	27.0	11	36.7
High	283	81.8	146	62.7	15	50.0
Dependent variable						
Prevention and control of behavior						
Low	24	6.9	8	3.4	4	13.3
Moderate	12	3.5	28	12.0	5	16.7
High	310	89.6	197	84.5	21	70.0

Footnote: Cutoff levels were categorized according to Bloom’s criteria: <60% = Low, 60–79% = Moderate, ≥80% = High. Different levels had different total scores; therefore, cutoffs were applied proportionally based on each level’s maximum score.

**Table 3 ijerph-22-01849-t003:** Factors association with COVID-19 prevention and control behaviors.

Factors	COVID-19 Preventive and Control Behaviors					
	Community Member (*n* = 346)		Volunteer (*n* = 233)		Government Officials (*n* = 30)	
	r	*p*-Value	r	*p*-Value	r	*p*-Value
Predisposing						
Perception	0.020	0.714	0.18	0.007 *	−0.17	0.363
Attitude	0.14	0.008 *	0.26	<0.001 *	−0.12	0.528
Reinforcing						
Social support	0.66	<0.001 *	0.38	<0.001 *	0.50	0.005 *
Participation	0.61	<0.001 *	0.10	0.143	0.47	0.008 *
Enabling						
Service	0.65	<0.001 *	0.16	0.015	−0.32	0.090

* Significant at α < 0.01 (2-tailed); Effect size interpretation (Cohen’s guideline [27] for Spearman’s rho): 0.10–0.29 = small effect (weak); 0.30–0.49 = medium effect (moderate); ≥0.50 = large effect (strong).

**Table 4 ijerph-22-01849-t004:** Summary of Qualitative Themes and Frequency of Mentions (POCCC Framework).

POCCC Component	Themes Identified	Example Codes	Frequency of Mentions (*n*)	Interpretation
P—Planning	Preparedness planning; simulation exercises; training and drills	“preparation before outbreak”, “simulation exercise”, “training personnel”	14	Highly emphasized across interviews
O—Organizing	Structure of provincial and district CDCU/EOC; formal operational system	“provincial committee”, “district EOC”, “incident command structure”	18	Most frequently mentioned component
C—Commanding	Unified incident commander; leadership capacity; chain of command	“district chief as commander”, “clear communication”	12	Leadership was a strong recurring theme
C—Coordinating	Cross-border coordination; inter-agency cooperation; informal networks	“TBC meetings”, “LINE coordination”, “informal channels”	16	Coordination challenges frequently noted
C—Controlling	Border control points; surveillance operations; community enforcement	“international checkpoints”, “natural routes”, “community monitoring”	15	Control operations were intensively described

## Data Availability

The original contributions presented in this study are included in the article. Further inquiries can be directed to the corresponding author.

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
