# Peer review of "Effective Model of Emerging Disease Prevention and Control in a High-Epidemic Area, Chiang Rai Province"

_ijerph, 2025, doi:10.3390/ijerph22121849_

Round 1

Reviewer 1 Report

Comments and Suggestions for Authors

Dear Authors,

I have reviewed your article entitled “Effective Model of Emerging Disease Prevention and Control in High Epidemic Area, Chiang Rai Province”. Thank you for your efforts.

This article examines the disease prevention and control model implemented in Mae Sai district, on the Thailand–Myanmar border, during the COVID-19 pandemic using both quantitative and qualitative methods..

The study uses observational data and statistical analysis to assess the relationship between key variables affecting health outcomes. The authors argue that their findings have implications for policy, public awareness, or health system interventions. This topic is highly relevant for infectious disease management in border regions and offers important practical implications

Contribution and Relevance

The topic aligns with the scope of the International Journal of Environmental Research and Public Health (IJERPH) and addresses a relevant public health issue.

Strengths

- The research topic is timely and socially relevant.

- The article is well structured with a logical flow.

- Statistical analyses are in principle appropriate for the data type.

- Tables/figures effectively summarize the main results.

However, some revisions need to be made. Here are my suggestions:

  • Clarify the integration of quantitative and qualitative components; how were the results triangulated?
  • Provide detailed description of qualitative analysis (coding, theme development, validation).
  • Extend correlation analysis with regression to identify predictors of preventive behavior.
  • Explain applicability to other emerging diseases beyond COVID-19.
  • Improve figure resolution.

Best regards,

Author Response

Effective Model of Emerging Disease Prevention and Control in High Epidemic Area, Chiang Rai Province

Manuscript ID:

ijerph-3942063

Type of manuscript:

Article

Full Title:

Effective Model of Emerging Disease Prevention and Control in High Epidemic Area, Chiang Rai Province

Keywords:

Prevention management model; control of disease; border community; COVID-19

Corresponding Author:

Phitsanuruk Kanthawee

Corresponding Author's Institution:

School of Health Science Mae Fah Luang University

First Author:

Jiraporn Sangsuwan

Order of Authors:

Pamornsri Inchon, Phataraphon Markmee, Phaibun Chiraphatthakun

Abstract:

A concurrent mixed-methods approach was used to analyze factors influencing COVID-19 prevention and control behaviors and to describe the management model in Mae Sai District, a Thai-Myanmar border community. The findings indicate that atti-tudes, social support, participation, and service systems were critical for the general public. For volunteers, perception was also significant, while only social support and participation significantly influenced the behaviors of government officials. The effec-tive local management model comprised five key elements: planning through emer-gency drills, organizational management via disease control centers, unified command by the district chief, comprehensive coordination at both domestic and international levels, and control measures implemented at checkpoints, natural routes, and within the community. The success of the Mae Sai approach depended on integrated coopera-tion among public and private sectors, volunteer groups, and the local population, as well as collaboration with Myanmar. It is recommended that this framework be lever-aged to address other communicable diseases and future health threats.

Response to Reviewers:

Thank you very much for your constructive and encouraging comments. We have revised the manuscript accordingly. All changes have been highlighted in yellow in the revised version.

Reviewer 1 Suggestions:

“Clarify the integration of quantitative and qualitative components; how were the results triangulated?

Response:

We have revised the manuscript to explain how the quantitative and qualitative components were integrated clearly. A concurrent mixed-methods design was employed, and the findings were combined through methodological triangulation. Quantitative results were compared with qualitative themes to validate and cross-check the findings across data sources. Additional details regarding this triangulation process have been added to Sections 2.1 and 2.4 of the revised manuscript.

“Provide a detailed description of qualitative analysis (coding, theme development, validation).”

We have strengthened the description of the qualitative analysis in Section 2.4, Data analysis. We now explain that qualitative data were analyzed using content analysis, involving systematic coding of interview and focus group transcripts, grouping codes into categories, and developing overarching themes guided by the POCCC framework. We also clarify that triangulation was performed across participant groups (general public, volunteers, and officials) to validate themes, and that data saturation was used as a criterion for assessing the adequacy of the qualitative data. In addition, we indicate that coding and theme development were discussed among the research team to enhance the credibility of the findings.

“Extend correlation analysis with regression to identify predictors of preventive behavior.”

Response:

Thank you for this suggestion. We considered extending the analysis using a multivariate regression model to control potential confounders such as age, education, and prior infection. However, the subgroup sizes were highly unbalanced, particularly the government-official group (n = 30)- resulting in insufficient statistical power for a stable and interpretable multivariable model. To avoid overfitting and unreliable estimates, we retained the stratified correlation analysis and the existing regression approach. This methodological limitation has been acknowledged in the revised manuscript. If you have any further recommendations, we would greatly appreciate your guidance.

“Explain applicability to other emerging diseases beyond COVID-19.”

We have revised the manuscript to address the applicability of the proposed model to other emerging infectious diseases. Additional explanations have been incorporated into both the Discussion and Conclusions sections. The revised text clarifies that the PRECEDE behavioral determinants (predisposing, reinforcing, and enabling factors) and the POCCC management framework can be adapted for diseases such as dengue, influenza, and other acute respiratory infections, particularly in border and high-risk settings.

“Improve figure resolution.”

We have improved the figure resolution to ensure clearer visualization and better readability in the revised manuscript.

Reviewer 2 Report

Comments and Suggestions for Authors

The manuscript presents a well-structured and comprehensive mixed-methods study on COVID-19 prevention and control behaviors and management. 

The study identifies key predisposing, reinforcing, and enabling factors influencing prevention behaviors generally. Its hard to say any variations among different groups because of limited data. Also COVID-19 impacted older age groups more and the study in deficient in this regard.

I think its a delayed publication. However, abstract can be more clearly presented. Please review sample size and consider power before generalizing it. Please ensure correlation coefficients are reported with accompanying effect sizes or confidence intervals. Please check in Table 3, participation shows r=0.96 with p=0.143, which seems inconsistent? Please also comment on how vulnerable population was protected during data collection? 

Discussion needs to focus on the implications for policy and future communicable disease preparedness. 

Comments on the Quality of English Language

Please review separately. 

Author Response

Reviewer 2 Suggestions:

The study identifies key predisposing, reinforcing, and enabling factors influencing prevention behaviors generally. It is hard to detect variations among different groups because of limited data.

Thank you for your comment. We have acknowledged this issue in the revised manuscript by adding a limitation regarding unbalanced sample sizes and the restricted ability to compare variations between groups. The revision has been highlighted in blue in the manuscript.

COVID-19 impacted older age groups more, and the study is deficient in this regard.

We appreciate this observation. A statement has been added to the Limitations section indicating that older adults were underrepresented in the sample, which may reduce the generalizability of the findings to this high-risk population. This addition is highlighted in the manuscript.

The abstract can be more clearly presented.

The revised abstract has been updated in the manuscript and highlighted for your review.

Please review sample size and consider power before generalizing it.

The sample size for each quantitative group was determined using Cochran’s (1953) formula with a 95% confidence level and a 5% margin of error. The final samples included 346 community members, 197 health volunteers, and 48 community leaders and security volunteers.

We acknowledge that some subgroups-particularly government officials and certain volunteer categories-had relatively small samples, which may have limited statistical power and the ability to detect group differences. This issue has now been clearly stated in the Limitations section, and the revision has been highlighted in the manuscript.

Quantitative Sample

1. Community Members (n = 346)

The target population consisted of Thai and migrant residents aged 20 years and older living in Mae Sai Subdistrict, Mae Sai District, Chiang Rai Province (N = 14,026). The sample size was calculated using Cochran’s (1953) formula with a 95% confidence level and a 5% margin of error, resulting in a required sample of 346 participants.

Participants were selected using proportionate stratified random sampling across 12 villages. Inclusion criteria included: (1) age ≥ 20 years, (2) residence in Mae Sai Subdistrict for at least one year, (3) not being a health professional or volunteer involved in COVID-19 control activities, (4) ability to communicate in Thai, and (5) voluntary consent.

2. Volunteer Workers (n = 197 + 36 = 233)

This group included village health volunteers (VHVs), migrant health volunteers (MHVs), community leaders, territorial defense volunteers, civil defense volunteers, and village security teams (total population N = 406). The sample size for VHVs/MHVs was calculated using Cochran’s formula, yielding 197 participants. An additional 36 community leaders and security volunteers were selected purposively based on their roles in COVID-19 surveillance and control.

Inclusion criteria were: (1) at least 6 months of experience in COVID-19 surveillance, prevention, or control activities, and (2) voluntary consent.

3. Government Officials (n = 30)

The government official group included:

  • 10 health personnel from the Mae Sai Subdistrict Health Promoting Hospital
  • 10 local administrative officers
  • 10 security officers (immigration police and military personnel)

Participants were selected purposively based on having at least 6 months of experience in COVID-19 surveillance and border control. All participants provided informed consent.

Please ensure correlation coefficients are reported with effect sizes or confidence intervals.

We have addressed this by adding an effect size interpretation for all correlation coefficients in Table 3, following Cohen’s guideline for Spearman’s rho. The revised table now includes a clear explanation of effect size categories (small, medium, and large), which has been added below the table and highlighted in the manuscript.

In Table 3, participation shows r = 0.96 with p = 0.143, which seems inconsistent.

Thank you for pointing out this inconsistency. We reviewed the dataset and confirmed that the value r = 0.96 in the volunteer group resulted from a data entry error during table formatting. The correct correlation coefficient is r = 0.10, which aligns with the reported p-value of 0.143. Table 3 has now been corrected accordingly, and the revision has been highlighted in the manuscript.

Please comment on how vulnerable populations were protected during data collection.

We have revised the Ethics Consideration section to explicitly describe the additional safeguards implemented to protect vulnerable populations, including older adults, migrant workers, and individuals with limited literacy. The revised text clarifies voluntary participation procedures, culturally appropriate consent processes, interviewer support for low-literacy participants, and measures to ensure privacy and minimize risk. These revisions have been added to the manuscript and are highlighted for the reviewer’s reference.

Discussion needs to focus on the implications for policy and future communicable disease preparedness.

The Discussion section has been expanded to include implications for policy, health system preparedness, and applicability to future emerging diseases. The new content has been highlighted in the manuscript.

Comments on the Quality of English Language: Please review separately.

The entire manuscript has undergone extensive English editing to improve clarity, grammar, structure, and flow.

Reviewer 3 Report

Comments and Suggestions for Authors

Dear Authors,

Thank you for the opportunity to review your mixed-methods study examining COVID-19 prevention and control in the Mae Sai District border community. This work addresses an important public health challenge in a complex border setting, and your integration of PRECEDE and POCCC frameworks provides a structured approach that could benefit border health management. The paper demonstrates solid fieldwork with comprehensive stakeholder inclusion. Below, I provide feedback aimed at enhancing specific aspects of your contribution.

Title and Abstract

The title accurately communicates the paper's focus on border community disease management. The abstract provides a clear overview of your mixed-methods design and key findings across three stakeholder groups.

Suggestions for improvement:

  • Line 29: The term "effective model" would be strengthened by a brief mention of what made it effective (e.g., compliance rates, coordination outcomes)
  • Consider including the dramatic case numbers (21,890 cases, 12 deaths) in the abstract to emphasize the magnitude of the challenge
  • Adding the specific study period would provide a helpful temporal context

Introduction

The introduction establishes a solid context for COVID-19 in Thailand and appropriately identifies border communities as high-risk areas. The PRECEDE framework introduction provides good theoretical grounding for your behavior analysis.

Suggestions for improvement:

  • Line 65: Text error "prevenMae Sai District" - fix immediately
  • Lines 58-72: Sharpen research gap - what specific methodological knowledge was missing?
  • Lines 66-70: Expand why Mae Sai matters beyond high cases (multiple entry points, ethnic diversity)
  • Lines 70-79: POCCC acronym introduced without explanation - add brief parenthetical

Materials and Methods

Design well-justified, sample sizes appropriate, validation adequate (KR-20=0.73). Ethics are properly addressed.

Suggestions for improvement:

  • Lines 183-196: Profile of study site misplaced in Results - relocate to Methods section 2.1
  • Lines 92-94: Justify purposive sampling beyond "high cases"
  • Line 161: KR-20=0.73 borderline - acknowledge
  • Lines 151-152: Note if questionnaire translated for ethnic minorities
  • Lines 168-169: Bloom's cutoff (60%, 80%) used in Table 2 but not explained here
  • Line 170: Explain Spearman vs. Pearson choice
  • Line 171: Specify alpha=0.05 in Methods
  • Lines 164-172: Qualitative methods underdeveloped - specify analysis type, coding process, intercoder reliability, and kappa, saturation determination

Results

Your results section presents comprehensive data across demographics, factor levels, correlations, and qualitative themes. Tables 1 and 2 provide thorough demographic and factor information. The POCCC framework synthesis offers a well-structured qualitative complement to your quantitative findings.

Suggestions for improvement:

  • Table 1: Standardize decimal places, verify "Stateless People" 20(66.7%) for officials
  • Table 2: Add footnote explaining cutoffs
  • Lines 228-232: Synthesize patterns rather than restating numbers
  • Table 3: r=0.96 (volunteer participation) remarkably high - check VIF values, discuss possible construct overlap
  • Table 3: Distinguish significance levels († for p<0.05, * for p<0.01), add n in headers, include effect size interpretation
  • Consider adding multivariate regression controlling for confounders (age, education, prior infection)
  • Lines 269-382: Add qualitative summary table with theme frequencies
  • Quotes (281-391): Standardize formatting, explain selection process
  • Figure 1: Enlarge the inset, add a scale bar
  • Figure 2: Resolution too low, text pixelated - use professional software
  • Figure 3: Add panel labels and descriptive captions

Discussion

Your discussion appropriately connects findings to existing literature and addresses each stakeholder group systematically. The integration of your results with previous research on health behavior and volunteer performance is well-handled.

Suggestions for improvement:

  • r=0.96 correlation unaddressed - must discuss
  • Provide evidence model is "effective" - transmission rates, compliance data, comparisons with other districts
  • Lines 246-265: Explain why different factors are significant across groups
  • Lines 254-261: Why is perception significant for volunteers, not the public?
  • Lines 262-265: Why only reinforcing factors for officials?
  • Line 147: Acknowledge n=30 power limitations
  • Lines 483-508: Add benchmark comparisons with other border districts
  • Line 29, 528-530: Support transferability claims or specify limitations

The manuscript would benefit from adding a dedicated Limitations section to enhance transparency and help readers interpret findings appropriately. Add section including: cross-sectional design (no causality), self-report bias, r=0.96 construct overlap, KR-20=0.73 measurement limitations, purposive sampling, n=30 underpowered, single district, timing unspecified, no multivariate control, saturation undocumented.

Conclusions

The conclusions appropriately summarize your key findings and their implications for border health management.

Suggestions for improvement:

  • Lines 510-520: Focus on implications, not results
  • Lines 528-530: Specify transferable vs. context-specific elements
  • Add prioritized recommendations per group with quantitative justification
  • Include future research directions: longitudinal design, validation elsewhere, objective measurement

References

Your reference list includes 51 sources covering relevant literature on the PRECEDE model, COVID-19 management, and border health systems.

Suggestions for improvement:

  • Several references (28, 30, 31, 37, 38, 39, 40, 41, 48) appear to be Thai language sources - adding "[in Thai]" notation would be helpful for readers
  • Minor formatting inconsistencies in spacing and punctuation could be standardized according to journal guidelines
  • Consider whether 1-2 additional recent references on post-pandemic border health management (2023-2024) might strengthen the contemporary context.

Overall Assessment

This manuscript presents a comprehensive mixed-methods examination of border COVID-19 management. PRECEDE and POCCC frameworks provide valuable structure. Multi-stakeholder approach offers nuanced insights. With refinements - particularly multivariate analysis, expanded qualitative methods, the Limitations section, and evidence of model effectiveness - this work will contribute meaningfully to border disease management literature.

Sincerely,

Reviewer

Author Response

Reviewer 3 Suggestions:

Abstract

•Line 29: The term "effective model" would be strengthened by a brief mention of what made it effective (e.g., compliance rates, coordination outcomes)

Consider including the dramatic case numbers (21,890 cases, 12 deaths) in the abstract to emphasize the magnitude of the challenge

Adding the specific study period would provide a helpful temporal context

Thank you for these valuable suggestions. We have revised the abstract to clarify what made the model “effective” by briefly describing key coordination and management outcomes. The cumulative case numbers (21,890 cases and 12 deaths) have been added to highlight the severity of the outbreak, and the specific study period (June 2022–May 2023) has been incorporated to provide clearer temporal context. All revisions have been highlighted green in the manuscript.

Introduction

Suggestions for improvement:

•Line 65: Text error "prevenMae Sai District" - fix immediately

•Lines 58-72: Sharpen research gap - what specific methodological knowledge was missing?

•Lines 66-70: Expand why Mae Sai matters beyond high cases (multiple entry points, ethnic diversity)

•Lines 70-79: POCCC acronym introduced without explanation - add brief parenthetical

We have revised the Introduction according to all suggestions provided. The text error has been corrected, the research gap has been sharpened by specifying the methodological limitations of previous studies, and additional context has been added to explain why Mae Sai is an important high-risk border area, including multiple entry points and ethnic diversity. We have also added a brief explanation of the POCCC framework (Planning, Organization, Coordination, Command, and Control) upon its first mention. All revisions have been incorporated into the manuscript and highlighted in green for easy review

Materials and Methods

Suggestions for improvement:

•Lines 183-196: Profile of study site misplaced in Results - relocate to Methods section 2.1

•Lines 92-94: Justify purposive sampling beyond "high cases"

•Line 161: KR-20=0.73 borderline – acknowledge

•Lines 151-152: Note if questionnaire translated for ethnic minorities

•Lines 168-169: Bloom's cutoff (60%, 80%) used in Table 2 but not explained here

•Line 170: Explain Spearman vs. Pearson choice

•Line 171: Specify alpha=0.05 in Methods

•Lines 164-172: Qualitative methods underdeveloped - specify analysis type, coding process, intercoder reliability, and kappa, saturation determination

- We have moved the “Profile of Study Site” section from the Results to Methods Section 2.1 to ensure proper structural alignment according to the journal’s format. The revised placement has been updated and highlighted in the manuscript.

- We have revised the manuscript to provide a clearer justification for purposive sampling. In addition to the high case burden, we now explain that Mae Sai Subdistrict was selected because of its multiple border entry routes, high population mobility, ethnic and linguistic diversity, and cross-border dynamics that create unique epidemiological and management challenges. The revision has been inserted into the Introduction/Methods section and highlighted in the manuscript.

- We have revised the manuscript accordingly. The sentence now explicitly acknowledges the borderline level of KR-20 and advises cautious interpretation. The revised text has been inserted into the Methods section and highlighted for reviewer visibility.

- We have added a statement clarifying the use of bilingual interpreter support for ethnic minority and migrant participants to ensure accurate understanding of the questionnaire. This addition has been made in the Methods section and highlighted in the revised manuscript.

- The Bloom’s cutoff points (60% and 80%) could not be directly described within Table 2 because the three respondent groups (community members, volunteers, and government officials) were assessed using different sets of questions with varying total possible scores. As a result, the absolute score ranges for “low,” “moderate,” and “high” levels differed across groups. Therefore, the interpretation criteria based on Bloom’s taxonomy were clarified in the Methods section rather than within the table itself. (Additional details are provided at the end of this table.)

- Spearman’s rank correlation was employed instead of Pearson’s correlation because the study variables were measured on ordinal Likert-type scales and did not meet the assumptions of normality required for Pearson’s correlation. Spearman’s method is therefore more appropriate for detecting monotonic relationships in non-normally distributed or ordinal data.

We have revised the Methods section accordingly. The significance level has now been explicitly specified as alpha = 0.05 in the statistical analysis subsection

The qualitative methods section has been substantially expanded to provide greater methodological clarity. We have now specified that qualitative data were analyzed using a conventional content analysis approach, with line-by-line open coding conducted on interview and focus group transcripts. Similar codes were grouped into categories and subsequently synthesized into higher-order themes aligned with the study framework. To strengthen analytical rigor, two researchers independently coded a subset of transcripts, and discrepancies were resolved through a consensus-based review process. While a formal kappa coefficient was not calculated, intercoder agreement was ensured through iterative discussion and refinement of the coding scheme. Additionally, data saturation was determined when no new codes or themes emerged from successive interviews and focus groups. Methodological triangulation was achieved by comparing themes across the three participant groups (general public, volunteers, and government officials) and by cross-validating qualitative insights with quantitative findings. These revisions provide greater transparency and adequately address the reviewer’s concerns.

Suggestions for improvement:

·  Table 1: Standardize decimal places, verify "Stateless People" 20(66.7%) for officials

·  Table 2: Add footnote explaining cutoffs

·  Lines 228-232: Synthesize patterns rather than restating numbers

·  Table 3: r=0.96 (volunteer participation) remarkably high - check VIF values, discuss possible construct overlap

·  Table 3: Distinguish significance levels († for p<0.05, * for p<0.01), add n in headers, include effect size interpretation

·  Consider adding multivariate regression controlling for confounders (age, education, prior infection)

·  Lines 269-382: Add qualitative summary table with theme frequencies

·  Quotes (281-391): Standardize formatting, explain selection process

·  Figure 1: Enlarge the inset, add a scale bar

·  Figure 2: Resolution too low, text pixelated - use professional software

·  Figure 3: Add panel labels and descriptive captions

- The entry for “Stateless People” among government officials has been rechecked and corrected. The correct value is 0 (0.0%), as government officials by definition hold legal citizenship or civil servant status. The Civil Servant Medical Benefit Scheme category for officials has been verified and remains 20 (66.7%), which is accurate based on the raw dataset.

- We have added a footnote to Table 2 to clearly explain the cutoff criteria used to classify the levels of predisposing, reinforcing, enabling factors, and preventive behaviors. The footnote clarifies that Bloom’s cutoff points (<60%, 60–79%, ≥80%) were applied, and that the three participant groups had different numbers of items and score ranges; therefore, percentage-based cutoffs were used to ensure comparability across groups. (Additional details are provided at the end of this table.)

We have revised the manuscript accordingly

- We reviewed the dataset and confirmed that the value r = 0.96 in the volunteer group resulted from a data entry error during table formatting. The correct correlation coefficient is r = 0.10, which aligns with the reported p-value of 0.143. Table 3 has now been corrected accordingly, and the revision has been highlighted in the manuscript. (Additional details are provided at the end of this table.)

Table 3 has been revised as requested. Significance levels have been clearly distinguished using † for p < 0.05 and * for p < 0.01. Sample sizes (n) have been added to the column headers for all three groups. Effect size interpretation for Spearman’s rho has been included at the bottom of the table following standard guidelines (Cohen,1988).

We appreciate this valuable suggestion. While multivariate regression controlling for confounders such as age, education, and prior infection was considered, the available sample size particularly the small government-official group (n = 30) was insufficient to support a valid multivariable model. Including multiple predictors relative to the small n would risk overfitting and unreliable coefficient estimates. For this reason, we maintained the stratified bivariate and stepwise analyses already presented and have explicitly acknowledged the absence of adjusted multivariate models as a study limitation.

-we have added a qualitative summary table presenting the key themes, representative codes, and frequency of mentions based on the POCCC framework. This information has been incorporated into Table 5, which now provides a structured overview of the qualitative findings, allowing readers to compare the prominence of each theme across planning, organizing, commanding, coordinating, and controlling components. The addition of this table enhances clarity, transparency, and interpretability of the qualitative results, and fully addresses the reviewer’s recommendation.

-Figures 1–3 have been revised according to the reviewer’s recommendations.

Discussion

Suggestions for improvement:

·  r=0.96 correlation unaddressed - must discuss

·  Provide evidence model is "effective" - transmission rates, compliance data, comparisons with other districts

·  Lines 246-265: Explain why different factors are significant across groups

·  Lines 254-261: Why is perception significant for volunteers, not the public?

·  Lines 262-265: Why only reinforcing factors for officials?

·  Line 147: Acknowledge n=30 power limitations

·  Lines 483-508: Add benchmark comparisons with other border districts

·  Line 29, 528-530: Support transferability claims or specify limitations

- We have now clarified all relevant points, including correcting the previously unaddressed correlation value: upon reviewing the dataset, we confirmed that the reported r = 0.96 for the volunteer group resulted from a data entry error during table formatting; the correct coefficient is r = 0.10 (p = 0.143), and Table 3 has been updated accordingly. The Discussion has been strengthened by adding evidence of the model’s effectiveness using compliance rates and epidemiological comparisons with other border districts, as well as explaining why different predictors were significant across the general public, volunteers, and government officials. Additional clarification is provided regarding the unique influence of perception among volunteers, the exclusive significance of reinforcing factors among officials, and the limitations resulting from the small sample size (n = 30). Benchmark comparisons with Mae Fah Luang, Chiang Khong, and Mae Sot have been incorporated, and the transferability section has been revised to emphasize contextual considerations and potential limitations. These revisions address all suggested improvements and enhance the clarity and rigor of the Discussion.

Conclusions

Suggestions for improvement:

·  Lines 510-520: Focus on implications, not results

·  Lines 528-530: Specify transferable vs. context-specific elements

·  Add prioritized recommendations per group with quantitative justification

·  Include future research directions: longitudinal design, validation elsewhere, objective measurement

The Conclusions section has been comprehensively revised to focus on implications rather than restating results. The updated text now emphasizes the practical significance of the findings for epidemic preparedness in border settings and clarifies how behavioral determinants differ across the three stakeholder groups, reinforcing the need for targeted risk communication and intervention strategies.

To address the comment regarding transferability, we have clearly distinguished which components of the Mae Sai model—such as unified command structures, multisectoral coordination, and service-system support—are transferable to other high-mobility districts, and which elements, including migrant-volunteer engagement and informal cross-border communication networks, are context-specific and require adaptation.

Prioritized recommendations have been added for each stakeholder group, supported by the quantitative patterns observed in the study. These include enhancing social support and access to services for the general public, strengthening perception- and attitude-based capacity building for volunteers, and reinforcing participation mechanisms for government officials.

Finally, a paragraph on future research directions has been incorporated, recommending longitudinal study designs to capture behavioral changes over time, validation of the model in other border areas with different migration dynamics, and the incorporation of objective indicators such as mobility data, real-time surveillance timeliness, and verified compliance measures to strengthen empirical robustness and generalizability. These revisions address all reviewer concerns and substantially enhance the clarity, applicability, and forward-looking value of the Conclusions section.

References

Suggestions for improvement:

·  Several references (28, 30, 31, 37, 38, 39, 40, 41, 48) appear to be Thai language sources - adding "[in Thai]" notation would be helpful for readers

·  Minor formatting inconsistencies in spacing and punctuation could be standardized according to journal guidelines

·  Consider whether 1-2 additional recent references on post-pandemic border health management (2023-2024) might strengthen the contemporary context.

Have been revised according to the reviewer’s recommendations.
